# Multi-Objective Optimization of Rear Guide Vane of Diagonal Flow Fan Based on Robustness Design Theory

**Shuiqing Zhou** [1,2,*], **Biao Xu** [1,2], **Laifa Lu** [1,2], **Weiya Jin** [1,2] and **Zijian Mao** [1,2]

1    College of Mechanical Engineering, Zhejiang University of Technology, Hangzhou 310023, China
2    Institute of Innovation Research of Shengzhou, Zhejiang University of Technology, Shengzhou 312451, China
*    Correspondence: zsqwh986@zjut.edu.cn

**Abstract:** As an essential fluid machine in national production, the optimal design of the performance of fans has long been a focus of research. In the format of small diagonal flow fans with rear guide vanes, there are too many design variables, and the design parameters interact with each other, resulting in a degradation of the fan's performance under actual operating conditions. Under realistic conditions, many uncertainties exist in the object of study and fluctuations in variables and parameters lead to a specific deviation of the optimization target from the design target. A method is needed for the robust and optimal design of angular flow fans under stochastic operating conditions of the design variables. In this paper, a multi-objective aerodynamic performance powerful design method is proposed using Pearson correlation analysis, combined with CFD calculation method, Latin hypercube experimental design and the Kriging agent model, to optimize the rear guide vane structural parameters of the diagonal flow fan under uncertain aerodynamic performance conditions, analyze the flow field characteristics before and after the modification, and select the optimal design model for the release. The test results show that the total pressure of the optimized fan is increased by 86 Pa, and the noise is reduced by 2.4 dB. The proposed optimization method is effective and can also be used to optimize the performance of other types of fans.

**Keywords:** diagonal flow fan; Pearson correlation analysis; kriging agent model; aerodynamic robustness design

## 1. Introduction

As one of the most common fluid machineries in modern society, the diagonal flow fan has the advantages of compact structure, easy installation, low noise, wide working range, etc. The airflow axially through the fan internal, converted to oblique flow, radial outflow, internal flow uniform, vortex loss is small, widely used in factories and buildings ventilation and air exchange.

The pneumatic performance of a fan is influenced by the external working environment, internal random disturbance fluctuations and other factors, which seriously affect the stability of its working performance. Commonly used optimized designs are designed under stable conditions such as a defined external environment and geometrical parameters. However, uncertainties play a crucial role in influencing the actual working state of a flow fan. Fluctuations in variables and parameters lead to a certain deviation of the optimized objectives from the design objectives. Uncertainty research, the essential idea of robust design, is a design method developed on the three-time design method proposed by the Japanese scholar Genichi Taguchi. Depending on the purpose of the design application, the optimization design problem can be solved by adjusting the design variables or the choice of design parameters to reduce the variance of the system output due to the design parameters.

This paper's current state of research is presented in terms of uncertainty analysis, robustness optimisation design and multi-objective optimisation, respectively, in the field of rotating machinery.

　　　Scholars at home and abroad have done relevant research in the field of rotating machinery, combining uncertainty analysis and optimization design directions. Panizza [1] compared uncertainty analysis methods in quantifying uncertainty in the aerodynamic performance of centrifugal impellers and showed that the polynomial chaos-based expansion method was able to obtain results similar to those of the Monte Carlo method while significantly reducing the computational effort. The effect of production and operational uncertainties on the aerodynamic performance (adiabatic efficiency and pressure ratio) was investigated, and the best robust design parameters were obtained by a stochastic optimization algorithm. Javed et al [2] used the Monte Carlo method to analyse the uncertainties in parameters such as the geometry of the inlet and outlet blade top clearance of a centrifugal compressor blade, to investigate the effect of uncertainties due to production and operation on aerodynamic performance (adiabatic efficiency and pressure ratio) and to give the best robust design parameters by means of a stochastic optimisation algorithm.Jue Fu et al. [3] programmatically implemented the non-embedded polynomial chaos (NIPC) method. They applied the NIPC method to the uncertainty analysis of the Rotor 36 to obtain the effect of geometric uncertainty in the leaf top clearance on the aerodynamic performance and leaf top flow field of an axial flow compressor. Emory [4] investigated the uncertainty quantification problem in the numerical simulation of impeller machinery; the uncertainty quantification method was used to assess the impact of uncertainty in each link on the aerodynamic performance and to find out the parameters that have the most significant impact on the aerodynamic performance.

　　　When an optimization problem has more than one conflicting objective function, the best solution for one objective cannot optimize the other objectives simultaneously or even lead to the degradation of different exact values; this type of problem is called a multi-objective optimization problem. More and more scholars at home and abroad have begun to adopt multi-objective optimization algorithms to solve practical engineering problems and have done much work on this. Kim [5] uses a hybrid multi-objective genetic algorithm combined with a response surface approach to optimize the design of axial fans using a full-flow channel and a single-flow channel model, respectively. Ding [6] takes the promotion of fan efficiency and pressure rise as the optimization objectives to carry out multiparameter and multi-objective optimization research. The pressure and velocity distributions in an optimal fan are uniforms, the internal flow separation is weakened, and the influence of tip leakage flow is reduced, improving the fan's energy performance.

　　　There is little research by related scholars on the robustness optimization design of rotating machines under uncertain conditions such as speed fluctuations and angle of attack during the operation. Figure 1 shows robust optimization, where changes in the optimization variables have a low fluctuation on the value of the optimization target. Uncertainty-based optimization makes the performance as good as possible on the one hand. It ensures that the sensitivity of the optimization target to uncertain optimization variables is low on the other [7]. Li Zhen [8] combined the Monte-Carlo method, Latin hypercube sampling method, non-embedded chaotic polynomial method and Hess-Smith surface element method to study the effects on the aerodynamic performance of wind turbine airfoils under uncertain conditions of wind speed and angle of attack of the incoming flow. Tang Xinzhi [9] quantified the degree of influence of speed uncertainty on the efficiency-pressure ratio of a centrifugal compressor, taking speed fluctuations as an example, and proposed a method for aerodynamic optimization and robust design of a centrifugal compressor based on uncertainty analysis methods, agent models, and multi-objective optimization algorithms.

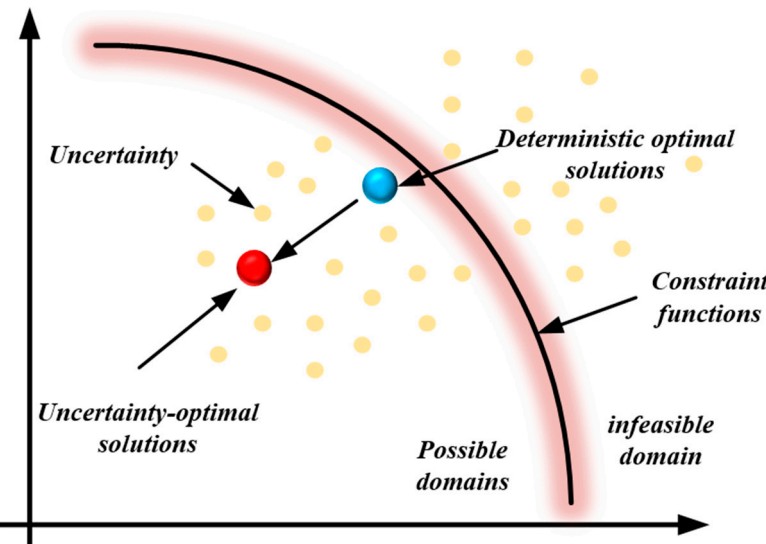

**Figure 1.** Robustness of constraints.

A diagonal flow fan is a rotating machine, and the internal flow field has strong three-dimensional characteristics. Based on the axial flow extension design of the flow fan, the impeller outlet spinning kinetic energy is relatively large. To make full use of this part of the energy, installing a guide vane at the rear of the impeller to convert the spinning kinetic energy into static pressure energy improves the static pressure efficiency of the fan. Domestic and foreign scholars use theoretical analysis, experimental research, numerical simulation and other methods on the rear guide vane, including the number of dynamic and static blade combinations, guide vane circumferential distribution law, dynamic and static blade clearance, guide vane installation angle and dynamic and static blade interference and a series of issues of research, these studies provide an essential reference basis for the design, optimization and theoretical analysis of the rear guide vane [10].

This paper examines the rear guide vane of a small diagonal flow fan, which is small, has a high speed and has a more complex internal flow. If the design is carried out under some single operating condition, too many design parameters are taken into account, leading to the deterioration of performance under non-designed operating conditions. Under realistic conditions, many uncertainties exist in the object of study and fluctuations in variables and parameters lead to a specific deviation of the optimization target from the design target. Therefore, the group introduces a variable-conditions, high-efficiency optimization strategy, choosing full pressure and sound pressure level as optimization targets to find the best design parameters for the rear guide vane.

## 2. Introduction of Research Subjects

In this paper, a small diagonal flow fan structure is studied, and the internal structure schematic diagram is shown in Figure 2. The system mainly consists of the flow collector, impeller, motor, rear guide vane, pressure expansion cylinder and other components, the structure of each component schematic diagram as shown in Figure 2, where the impeller is the only power original, choose 25 °C dry air as the flow medium. The design flow rate of the diagonal flow fan in the paper is $Q_v = 22$ m$^3$/min, the number of moving blades $Z_1 = 10$, the number of rear guide blades $Z_2 = 9$, the speed $n = 2250$ rpm, the maximum static pressure $P_s = 525$ Pa, the total force efficiency = 36.7%, the sound pressure level SPL = 62.1 dB. A structure diagram of each component is shown in Figure 3.

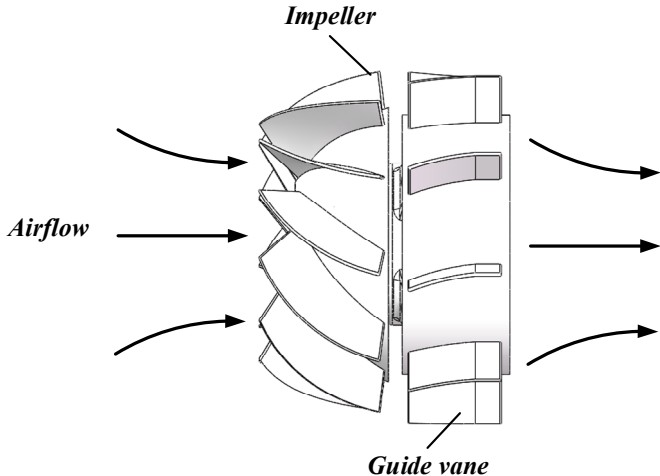

**Figure 2.** A structure diagram of a small diagonal flow fan.

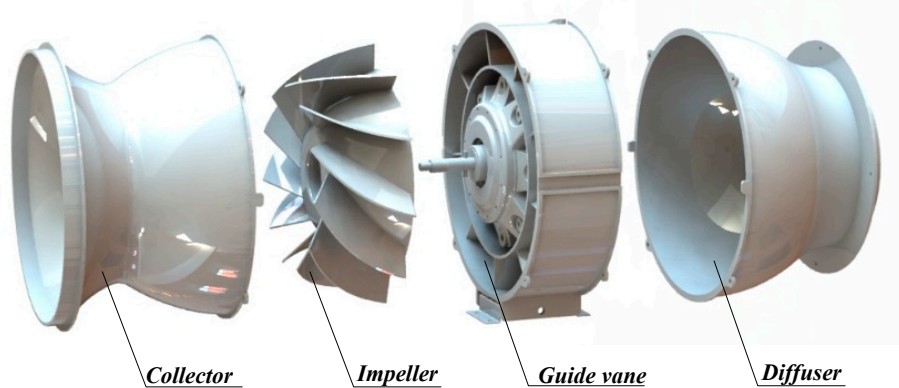

**Figure 3.** A structure diagram of each component.

## 3. Numerical Simulation Calculation Method and Experimental Validation

### 3.1. Computational Model and Meshing

The physical 3D model was simplified and imported into ICEM CFD and divided into an inlet fluid domain, a moving vane fluid domain, a rear guide vane fluid domain and an outlet fluid domain according to the actual airflow area, as shown in Figure 4 below. To accurately predict the actual operating conditions of the flow fan, the length of the inlet and outlet basins was extended to five times the existing channel and outlet diameter size.

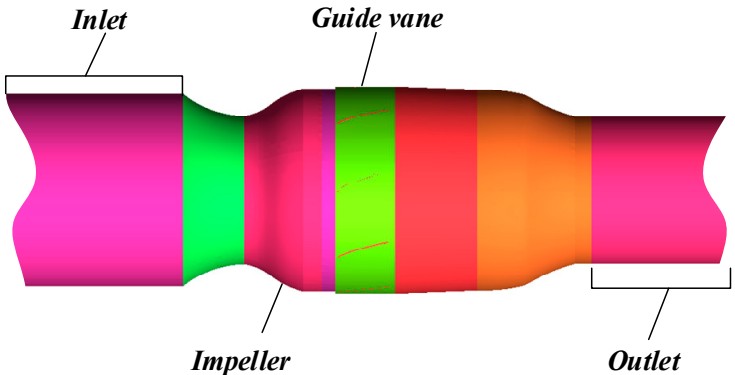

**Figure 4.** The calculation domain division.

In the numerical simulation calculation, to save calculation time and resources and to choose a suitable grid model to ensure accurate results, this paper adopts a hexahedral structured grid for the diagonal flow fan motor vane and rear guide vane. An unstructured

grid is used for irregular structures such as inlet and outlet, flow collectors and diffusers. Figure 5 shows the grid division of each part.

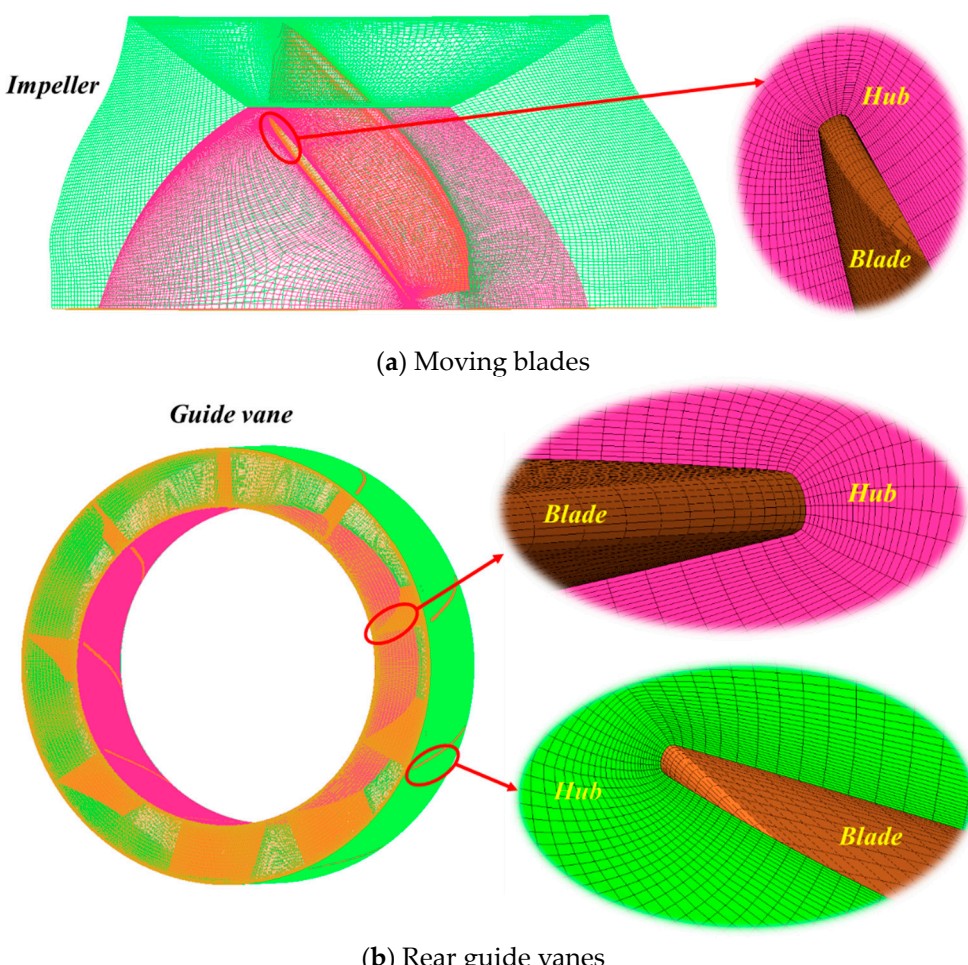

(**a**) Moving blades

(**b**) Rear guide vanes

**Figure 5.** A computational grid model.

The internal circulation fluid domain of the flow fan has meshed, and the main components are the moving vane and the rear guide vane. TurboGrid is a professional turbine lobe grid channel meshing software that allows high-quality structured meshing of complex-shaped blades to improve the accuracy of the calculation. The drawn mesh file is imported into the ICEM CFD for analysis.

The moving vane area grid is shown in Figure 5a, and the rear guide vane area grid is shown in Figure 5b; the inlet and outlet fluid domains, which are more regular in shape, are divided into O-Block grids. Due to their complex and irregular structure, non-structural meshing is used for the rest of the overflow components. When dividing the mesh, the first node is required to be arranged in the region of the viscous boundary layer so that it is $y^+ \leq 5$, where the empirical formula for $y^+$ is [11]:

$$Y_{wall} = 6 \left( \frac{V_{ref}}{v} \right)^{-\frac{7}{8}} \left( \frac{L_{ref}}{2} \right)^{\frac{1}{8}} y^+ \tag{1}$$

$V_{ref}$ is the reference velocity, m/s; $L_{ref}$ is the reference length, m; $v$ is the kinematic viscosity of the fluid, m$^2$/s; $y^+$ is a dimensionless parameter. According to the empirical formula, the height of the first layer of this boundary layer grid should be less than 0.65 mm.

### 3.2. Boundary Conditions

According to the calculation model and meshing in Section 3.1, ANSYS FLUENT 19.2 software was used for the steady-state numerical calculations. The numerical simulation of the diagonal flow fan uses a steady-state (Steady) method based on a Pressure-Based solver, which uses sequential solving to set the variables—setting the dynamic lobe fluid domain as the rotational domain, the outlet cross section in the outlet fluid domain as the flow monitoring surface, and the inlet and outlet boundary conditions as the pressure-inlet and pressure-outlet. The turbulence modeling uses the Realizable k-$\varepsilon$ eddy viscosity model, which is highly accurate in calculating rotational uniform shear flows, boundary layer flows and separation flows. The SIMPLE algorithm was used to calculate the flow field, and the transport equations were discretized by the finite volume method. The second-order accurate discretisation is chosen for individual terms and the second-order upwind is selected for the convection terms. The root means square (RMS) value of the residuals of the control equation is less than $10^{-5}$ to guarantee the accuracy of the numerical calculation.

Figure 6 shows the grid number's influence on the numerical simulation results of the diagonal flow fan. This fan's total pressure and flow rate stop changing after the grid number exceeds 3 million. The calculation grid number is finally determined to be 3 million after considering the calculation accuracy and time.

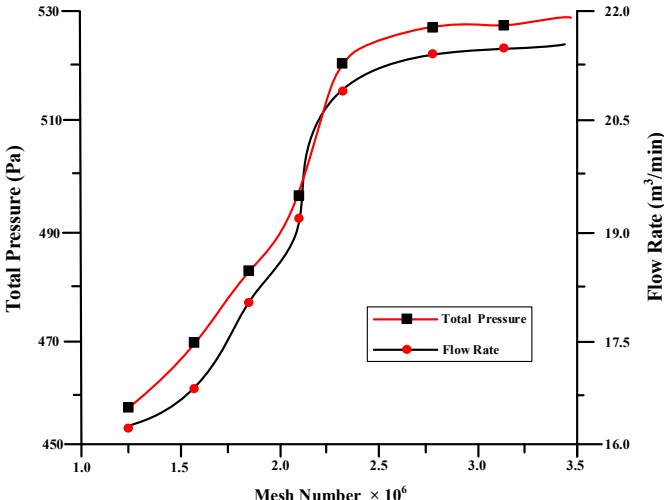

**Figure 6.** The grid independence verification.

## 4. Multi-Objective Geometric Uncertainty Pneumatic Robustness Optimization

Numerical simulation techniques, such as CFD, have become indispensable tools in scientific research and the analysis and design of practical engineering problems. However, existing solvers are deterministic, i.e., input specific boundary conditions or given geometrical parameters unique to a particular solution. However, under realistic conditions, the object of study is subject to several uncertainties, leading to uncertain boundary conditions, geometrical delays and so on. The presence of these uncertainties can lead to deviations between the simulation results and the results under actual operating conditions [12].

### 4.1. Determining Variable Parameters

Installing guide vanes on the fan is a suitable solution to encountering air swirling or losses in a fan system. Reducing the swirl energy losses in a fan system can improve the total efficiency of the system [13]. Before and after the rear guide vane reshaping, blade shape and installation angle schematic diagram as shown in Figure 7, guide vane geometric parameters and inlet and outlet velocity schematic diagram, as shown in Figure 7, where the guide vane installation angle is the guide vane bending angle, $C$ is the guide vane chord length, $b$ is the guide vane thickness, $V_1$ is the inlet absolute axial velocity, $V_2$ is the outlet

absolute velocity, $i$ is the angle of attack, $\delta$ is the deviation angle, $\boldsymbol{a}_1$ is the inflow angle, $\boldsymbol{a}_2$ is the outflow angle.

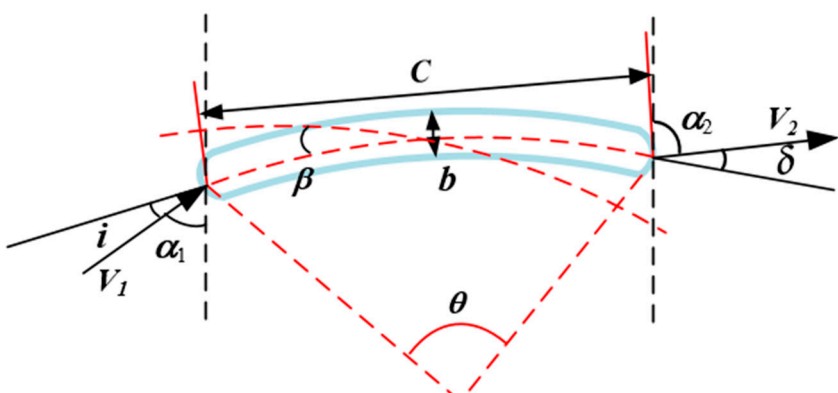

**Figure 7.** A schematic diagram of guide vane geometric parameters.

*4.2. Pearson Correlation Coefficient*

Correlation analysis is the analysis of two or more elements of a variable that are correlated to measure the closeness of the correlation between the two variables and the linearity of the relationship between the elements. The small diagonal flow fan studied in this paper involve many study parameters. If all of them are considered design variables for optimal design, the number of sample spaces is too large. It is necessary to reduce the sample size by removing the less influential parameters according to their degree of influence on the performance of the fans [14].

The Pearson correlation coefficient is the most common method used when studying the correlation between two variables and requires the calculation of the following two coefficients, the sample correlation coefficient (r) and the overall correlation coefficient (ρ).

The sample correlation coefficient, replacing the overall covariance and standard deviation with the sample's covariance and standard deviation calculated.

Pearson correlation coefficients range from [−1, +1], with less than 0 representing a negative correlation, greater than 0 representing a positive correlation, and equal to 0 representing no correlation. The closer the correlation coefficient is to 0, the weaker the correlation is; the closer it is to −1 or +1, the stronger the correlation is.

In this paper, Pearson correlation analysis was performed with Spss software to detect the degree of linear correlation between several parameter variables of the rear guide vane of the diagonal flow fan in order to avoid the influence of the design sample size on the judgement of the degree of correlation between the optimization variables and the optimization objective.

Figure 8 shows the correlation between the optimization variables of the rear guide vane and the optimization objective, ranking the optimization variables in order of the magnitude of the correlation coefficient as speed, chord length, mounting angle, bending angle and guide vane thickness [15].

Based on the 60 samples in Table 1, the optimal Latin hypercube experimental design was used. The selected sampling points were more complete in spatial distribution, and there was no problem of missing sample points locally. The final sampling ratio was determined through numerous trainings to ensure the accuracy of the proxy model, as shown in Figure 9.

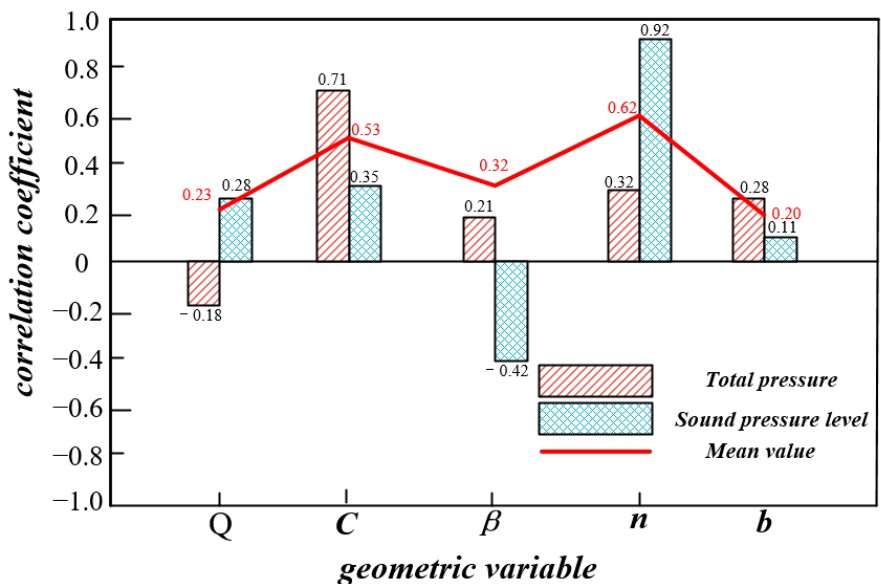

**Figure 8.** The correlation between optimization variables and optimization objectives.

**Table 1.** The value range of design parameters.

| Name | Design Parameters | Upper Limit | Lower Limit |
|---|---|---|---|
| Bend angle | $\theta$ | 85° | 76° |
| Length of string | $C$ | 84 mm | 62 mm |
| Mounting angle | $\beta$ | 19° | 10° |
| Design speed | $n$ | 2300 rpm | 2000 rpm |
| Guide leaf thickness | $b$ | 1 mm | 3 mm |

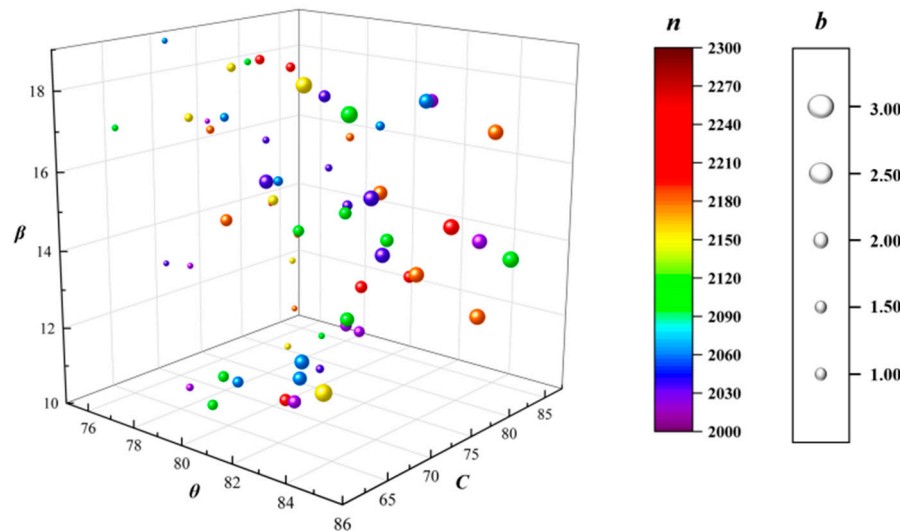

**Figure 9.** A sampling of 60 original samples.

The response value sound pressure level and full pressure were obtained with numerical calculation and Pearson correlation analysis. The kriging proxy model is established with the guide vane installation angle as the angle, the guide vane bending angle as the angle, C as the guide vane chord length and the speed $n$ as the optimization variables, and the sound pressure level SPL and full pressure P of the diagonal flow fan as the optimization targets. The $R_2$ value ($0 \leq R_2 \leq 1$) is above 0.9, and the predicted value matches the calculated value.

Where the design speed is 2250 rpm, only increasing or decreasing the speed is not in line with the original intention of optimizing the structural design of the fan, and the guide vane thickness has the most negligible effect on the optimization objective. Therefore, the chord length, mounting angle and bending angle are selected as multi-objective optimization variables for the diagonal flow fan.

With the optimization objective of minimizing sound pressure level and maximizing full pressure, the mathematical model is developed as follows. The range of values for the optimization variables is based on the design experience of the study fan and the data from the actual operating conditions.

$$\begin{cases} \max(P), \min(SPL) \\ x = (\beta, \theta, C, n) \\ s.t. \beta \in [10, 19]; C \in [62, 84] \\ \theta \in [76, 85]; n \in [2000, 2300] \end{cases} \quad (2)$$

Figure 10 shows the Pareto front solution to the problem, and it can be seen that there is a small gap between the solution sets. The optimization work should not only focus on static pressure increase, but the diagonal flow fan also needs to focus on noise. This paper uses a solution that meets the maximum value of 0.5 SPL + 0.5 P and selects the optimum from a sample of three optimized impellers to maximize the integration of performance indicators.

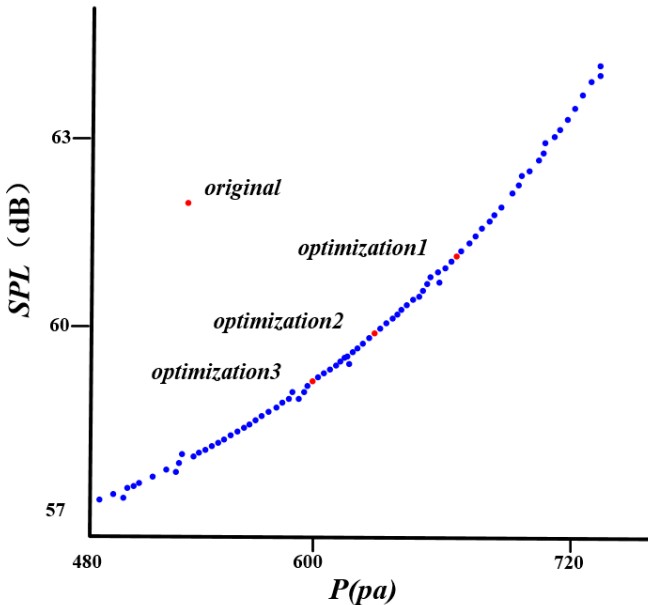

**Figure 10.** Pareto frontier for the optimization objective.

Ultimately, optimization 3 was chosen as the optimal solution. The structural parameters corresponding to the sample points of the optimal solution are listed in Table 2, where the guide vane bending angle, chord length and mounting angle become larger.

**Table 2.** A comparison between prototype and optimization results.

| Name | Bend Angle $\theta$ (°) | Length of String $C$ (mm) | Mounting Angle $\beta$ (°) | Design Speed $n$ (rpm) |
|---|---|---|---|---|
| original | 78 | 62 | 10 | 2250 |
| optimization | 84 | 81 | 16 | 2250 |

Compared to the original guide vane, the guide vane is optimized for a complete pressure increase of 86 Pa and a noise reduction of 2.4 dB.

The Non-Dominated Sorting Genetic Algorithm optimizes the mathematical optimization model with Elite Strategy (NSGA-II), a fast non-dominated sorting algorithm that,

on the one hand, reduces computational complexity and, on the other hand, effectively expands the sample dataset as the locally superior individuals are retained [16]. The algorithm introduces the concept of "crowding" in the global comparison so that individuals in the quasi-pareto domain can even be extended to the whole Pareto domain, thus ensuring the retention of the best populations, optimizing processes as shown in Figure 11 [17].

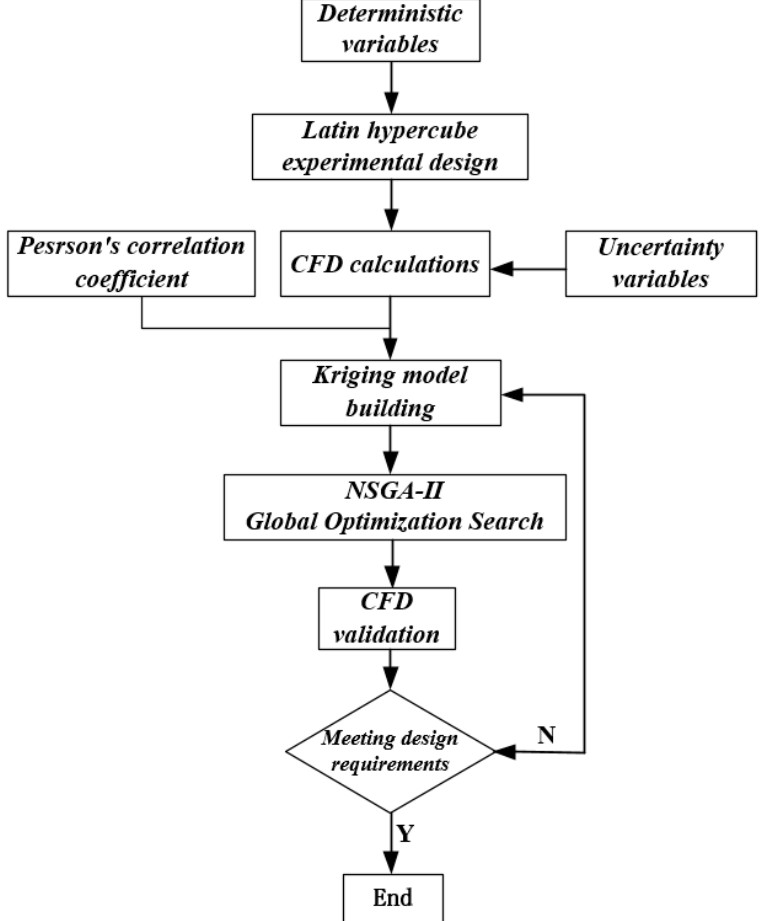

**Figure 11.** An optimization flow chart.

The algorithm is terminated by setting the maximum number of iterations and checking the convergence results of the population at each generation. The NSGA-II algorithm control parameters are designed as follows: the number of populations is 100, the maximum number of iterations is 1000, the probability of variation is 0.7, and the probability of crossover is 0.3.

## 5. Analysis of Results

A comparison of the specific dimensions of the optimum and the original guide vane is shown in Figure 12.

The model of the guide vane before and after optimization was prototyped and tested experimentally. The overall assembly entity of the fan is shown in Figure 13a, and the models before and after the modification are shown in Figure 13b,c, respectively.

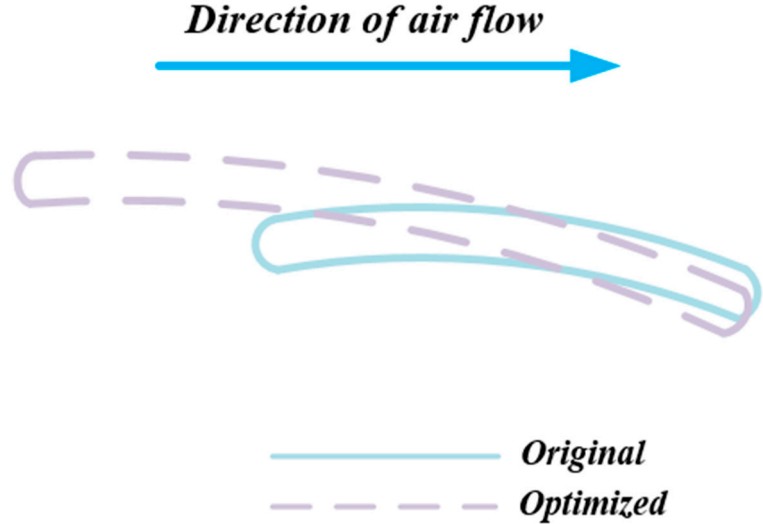

**Figure 12.** A before and after comparison optimization of the rear guide vane.

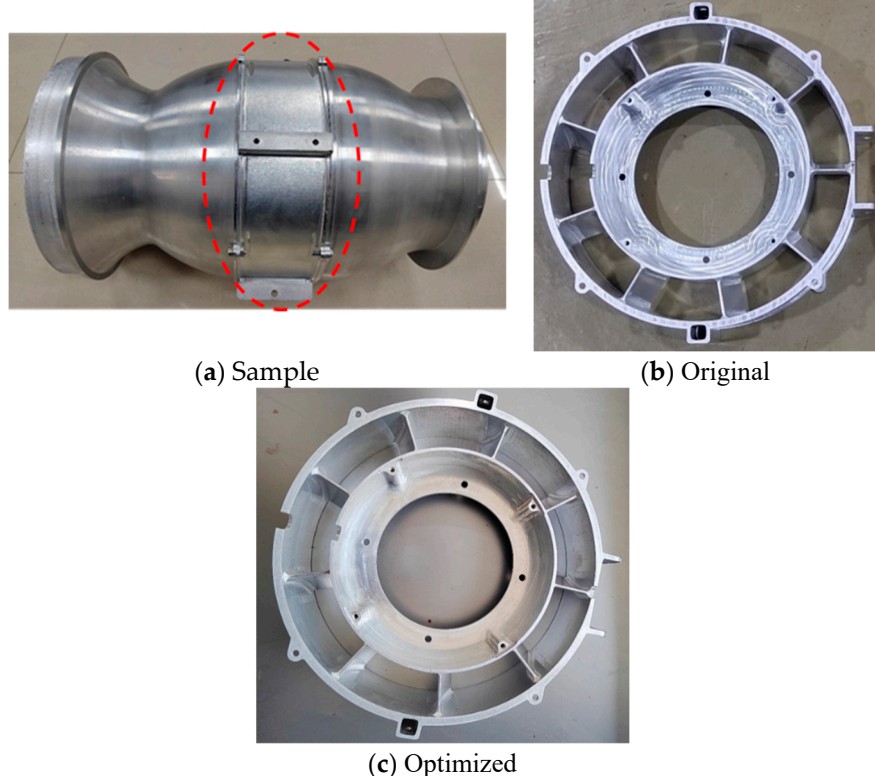

**Figure 13.** A sample comparison.

### 5.1. Analysis of Test Results

In order to verify the accuracy of the numerical simulation method during the retrofitting process, the aerodynamic characteristics of the prototype fans were tested. By the testing requirements of the current national standards GB/T1236-2000, The test apparatus is a type B external range hood air performance test apparatus, see Figure 14 for a schematic diagram of the test apparatus, see Figure 15 for test trials.

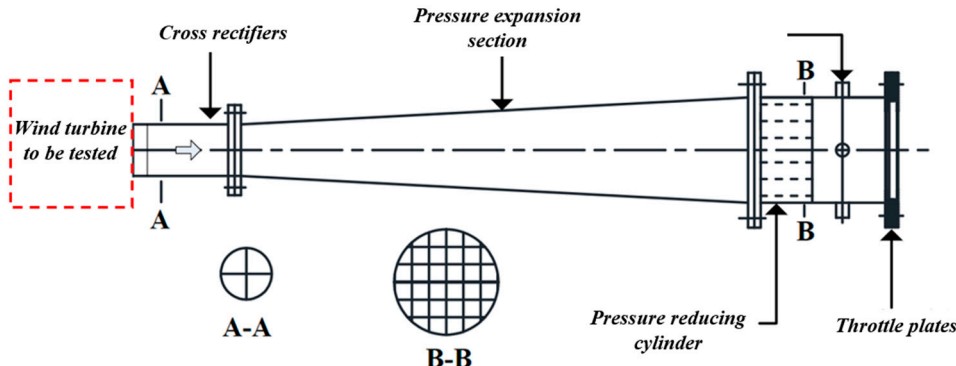

**Figure 14.** A schematic diagram of the ventilator performance test device.

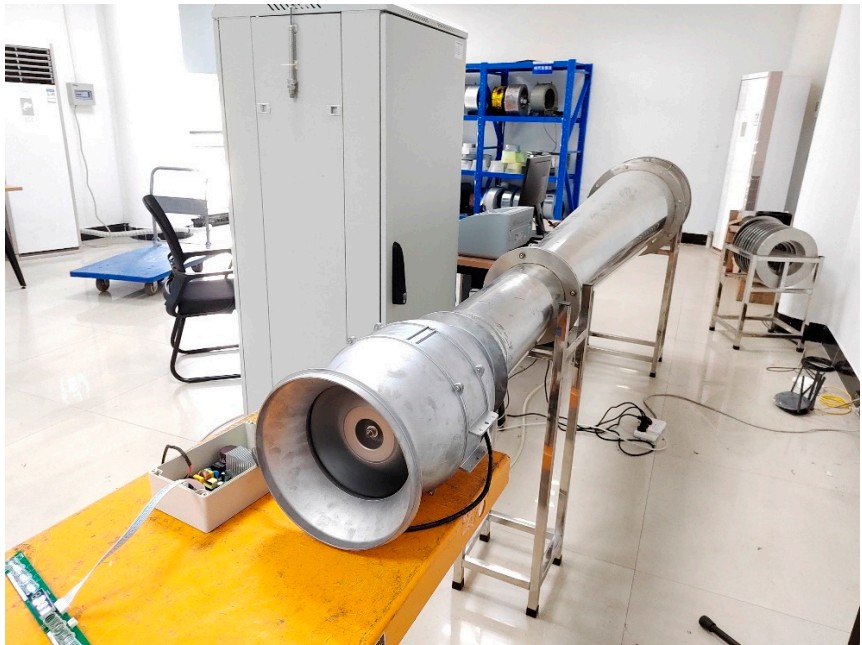

**Figure 15.** The aerodynamic performance test device.

The fan studied in this paper is a general form of inclined flow fan. In order to study the actual operating conditions of the fan, noise tests of A sound level and sound pressure level have to be carried out according to the National Standard of the People's Republic of China "GB/T 2888-2008 Methods of measuring noise of fans and Roots blowers". In order to obtain the distribution of aerodynamic noise along the circumference and at the outlet, it is necessary to measure the radiated noise at the outlet of the fan, as shown in Figure 16. An average of 12 measurement points are arranged around the circumference of the turbine, with the line from each measurement point to the noise source at 45° to the ground and the straight-line distance between the measurement points and the noise source at 1 m, as shown in Figure 17. As only four test probes were available under the experimental conditions, four were arranged in three experiments simultaneously, and 12 noise values were measured [18].

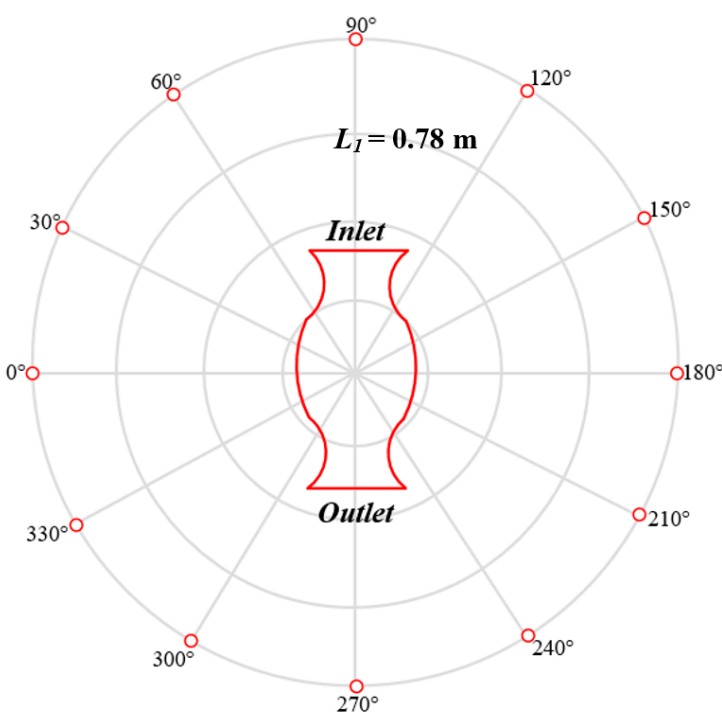

**Figure 16.** A noise test diagram.

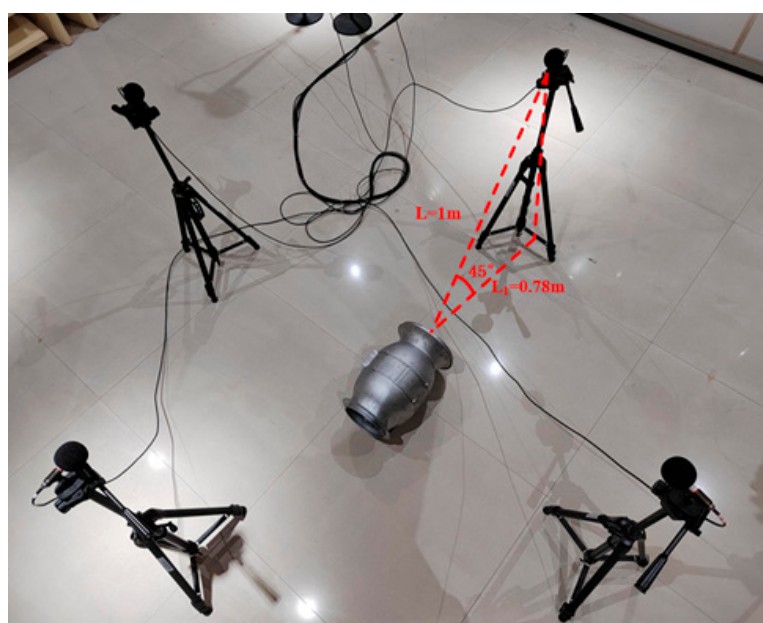

**Figure 17.** The noise performance testing.

### 5.2. Numerical Calculation Analysis before and after Optimization

To gain a deeper understanding of the uncertainty factor, the mechanism of influence on the overall aerodynamic performance of the fan, as well as the impact of the changes in the fan performance before and after the optimization of the rear guide vane of the inclined flow fan, the following study of the uncertainty flow phenomenon inside the rear guide vane is carried out.

For a more transparent analysis and comparison, a 0.5 blade height section of the rear guide vane is selected for investigation. The velocity distribution, flow field and turbulent energy distribution in the fan before and after optimization are analyzed separately at the operating design point.

As Figure 18 above shows the 3D surface plot of the velocity distribution at the 0.5 lobe height section before optimization, it can be seen from the plot that the velocity distribution of the section is uneven, and the velocity fluctuation difference is significant; there is a specific abrupt change phenomenon, apparent velocity distribution.

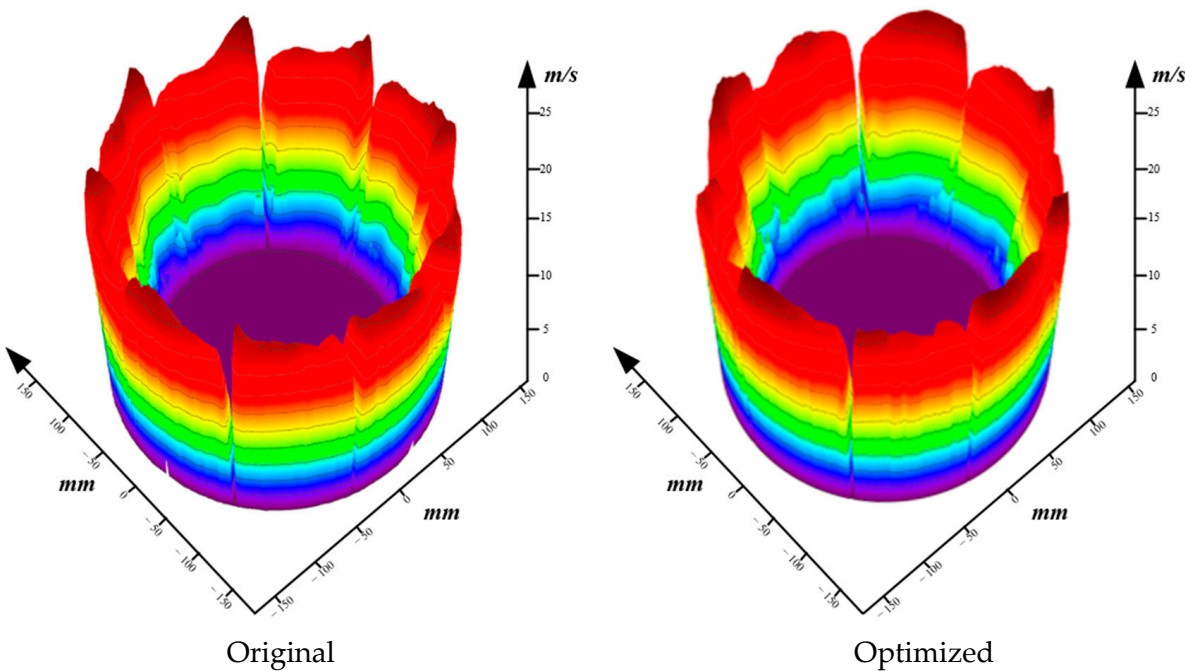

Original                                                                          Optimized

**Figure 18.** The velocity distribution of the cross-section before and after optimization.

Using the Turbo module in the CFD-POST software, the blades are expanded into the fluid domains of the moving and guide vanes. As shown in Figure 19, the 0.5 leaf height cross-section (blue face) of the moving and guide vanes is expanded into a 2D plane.

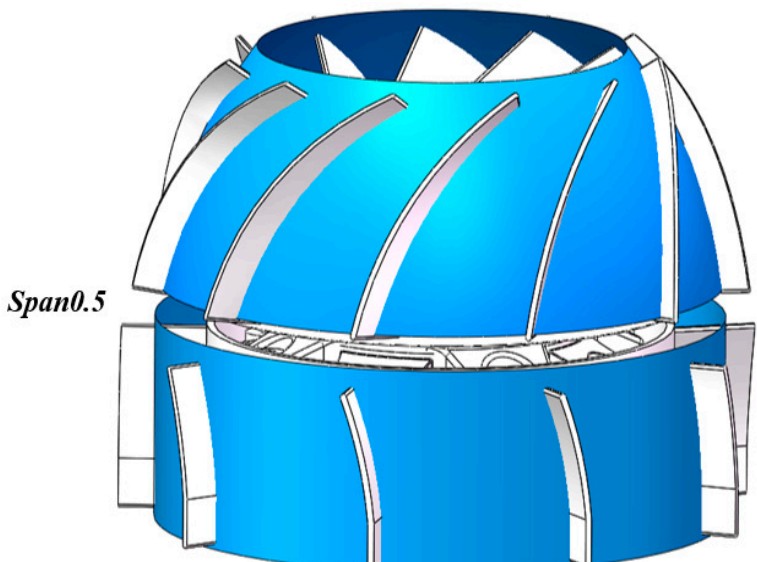

**Figure 19.** A leaf expansion diagram.

As shown in Figure 20, the velocity distribution around the guide vane is at 0.5 leaf height before and after optimization. The flow capacity of the guide vane area is indirectly reflected by the velocity distribution in this area, the velocity distribution gradient between

the guide vane before the modification is uneven, and the velocity difference between the pressure surface and the suction surface is significant. After the modification, the pressure surface forms a larger high-speed zone, which enhances the circulation capacity of the guide vane section, and the flow velocity increases significantly at the guide vane outlet, improving the efficiency of the guide vane.

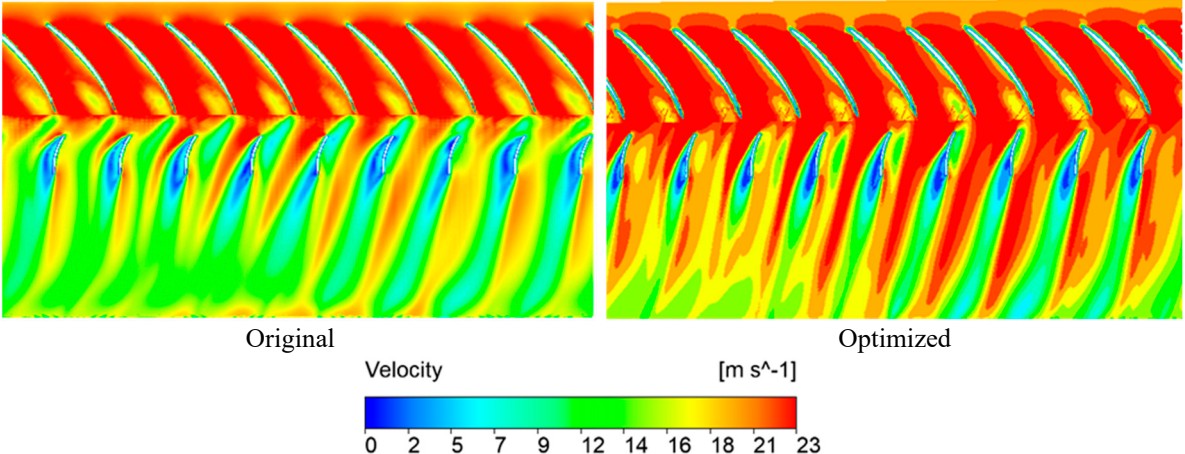

**Figure 20.** The velocity distribution at 0.5 blade height before and after optimization.

Figure 21 above shows the flow distribution around the guide vane at 0.5 leaf height before and after optimization. Before optimization, the negative angle of attack of the airflow is more prominent, and the pressure surface of the guide vane is prone to the backflow phenomenon, causing detachment loss and reducing the effective circulation area. The increase in the angle of attack of the surface layer into the flow is prone to the phenomenon of airflow separation; the formation of the blockage zone inhibits the flow instability of the mainstream inside the fan. After the blade transformation, the bending angle is reduced on the incoming flow of the dynamic lobe, playing a slowing role, effectively curbing the phenomenon of reverse flow and reducing the flow loss of the guide lobe.

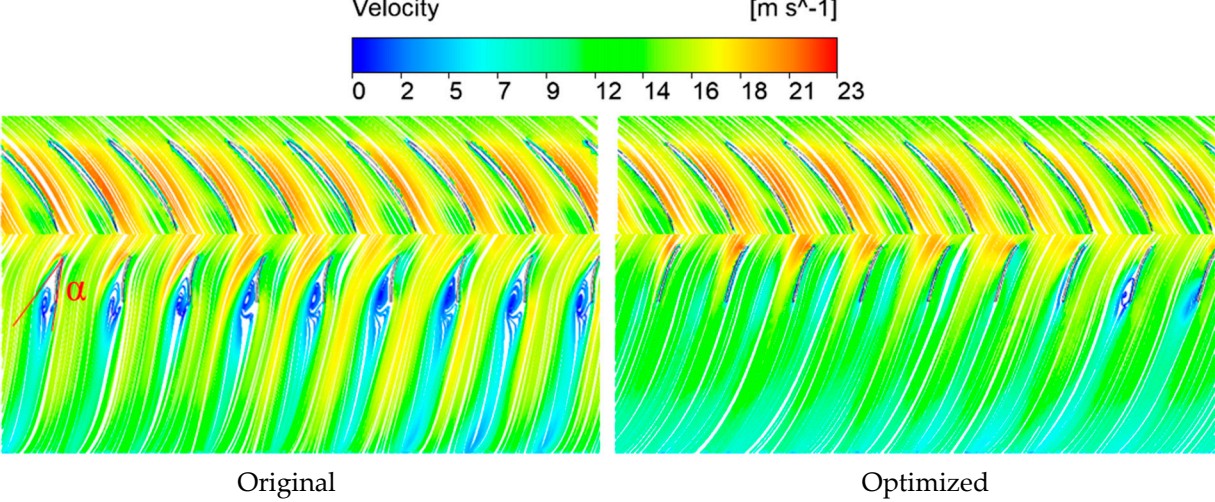

**Figure 21.** The streamline distribution at 0.5 blade height before and after optimization.

The modification scheme in this paper confirms the conclusions of Emmons [19], shown in Figure 22, who suggests that the onset of stall is related to the blade inlet impulse angle and explains the mechanism by which rotational stall propagates between the guide vanes. As the impeller flow decreases and the guide vane inlet impulse angle increases, any slight asymmetry in the flow path structure will first cause flow separation in one or

more fluid domains and further block the fluid domains. The incoming flow will change direction towards the adjacent impeller. The incoming flow will change direction towards the neighbouring impellers, which corresponds to a change in the direction of the incoming flow of the neighbouring impellers. For the upstream neighbouring lobe channel, the angle of entry will increase, and the flow separation in this lobe channel will increase; conversely, the angle of entry impulse in the downstream neighbouring lobe channel will decrease, and the separation on the suction side of the lobe channel will be somewhat suppressed so that the stalled group achieves a reverse propagation on the blade row.

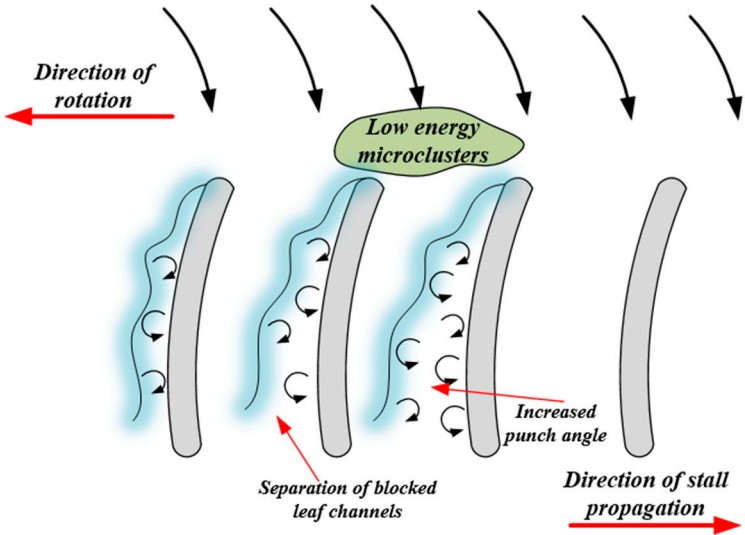

**Figure 22.** The formation of flow instability.

Turbulent energy can characterize the intensity of fluid turbulence and the degree of turbulent pulsation. Its magnitude and spatial inhomogeneity also reflect, to some extent, the magnitude and extent of pulsation dispersion and viscous dissipation losses [20]. In order to investigate the effect of the structural design of the rear guide vane of the inclined flow fan on the internal flow characteristics of the fan, the turbulent energy distribution at 0.5 lobe height before and after optimization at the same flow condition is shown in Figure 23. The internal turbulent energy distribution in the original rear guide vane area is more significant than that in the moving vane section. However, the internal turbulent energy distribution in the rear guide vane area is sharply reduced after the change.

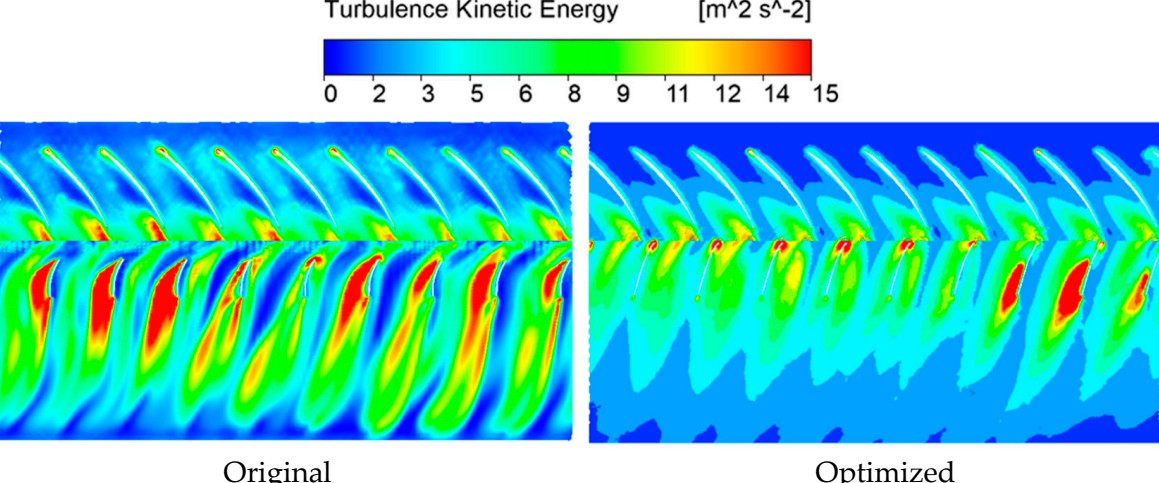

Original                                        Optimized

**Figure 23.** The turbulent kinetic energy distribution at 0.5 blade height before and after optimization.

The wavelet transform derives significant fluctuating patterns in the time series by decomposing the time series into the time-frequency domain, i.e., the periodically varying dynamics and the temporal pattern of the periodically varying dynamics. After wavelet variation processing, weak local features are brought to the before [21].

As shown above, wavelet analysis decomposes a signal into a series of superimposed functions derived from a mother wavelet function through translation and scale scaling. This irregular wavelet function can be used to approximate the sharply varying parts of a non-stationary signal. It can also be used to approximate discrete, discontinuous signals with local characteristics, thus reflecting more realistically the changes in the original signal on a given time scale. This local nature of wavelet analysis makes it an effective tool for quantifying non-stationary, discontinuous time series. The definition of the continuous wavelet transform can be expressed as [22,23]:

$$W_f(a,b) = \frac{1}{\sqrt{|a|}} \int_R f(t)\overline{\varphi}\left(\frac{t-b}{a}\right) dt \tag{3}$$

where $\varphi$ represents the mother function of the wavelet function, $a$ represents the scale factor, and $b$ represents the translation factor.

The more wavelet function is used as the basis function for the transformation, and the order of the wavelet function is 5, cmor-5 wavelet, and the wavelet processing of the pressure pulsation signal of the modified guide vane is developed, as shown in Figure 24. The wavelet signal consists of a significant surge at the fundamental and octave frequencies, while there are also more apparent fluctuation values around 120 Hz.

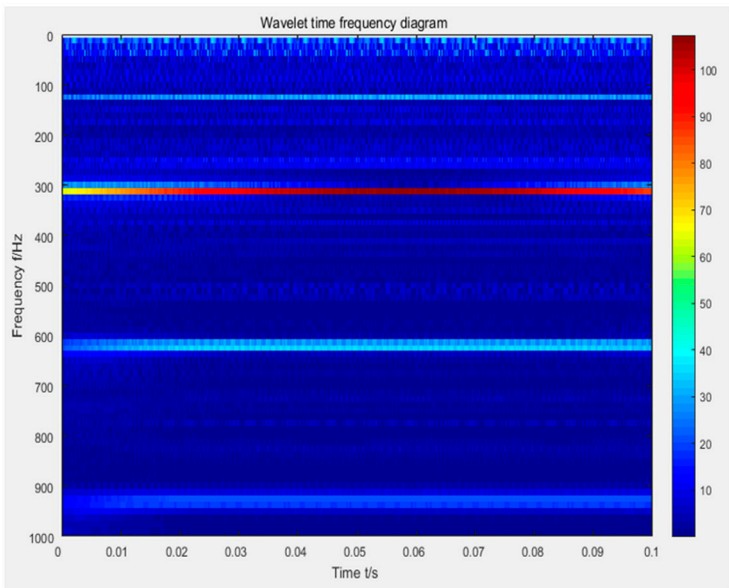

**Figure 24.** A wavelet spectrum analysis of a modified guide vane.

Fan noise is directional and needs to be based on the experimental results before and after the fan outlet and circumferential directional aerodynamic noise direction to characterize the overall fan aerodynamic noise as a way to judge the effect of the entire fan modification [24]. As shown in Figure 25, the fan inlet and outlet direction noise are the largest, the overall "8" type, the fan circumferential plane of the aerodynamic noise has dipole characteristics, and the overall noise in the circumferential plane has been reduced.

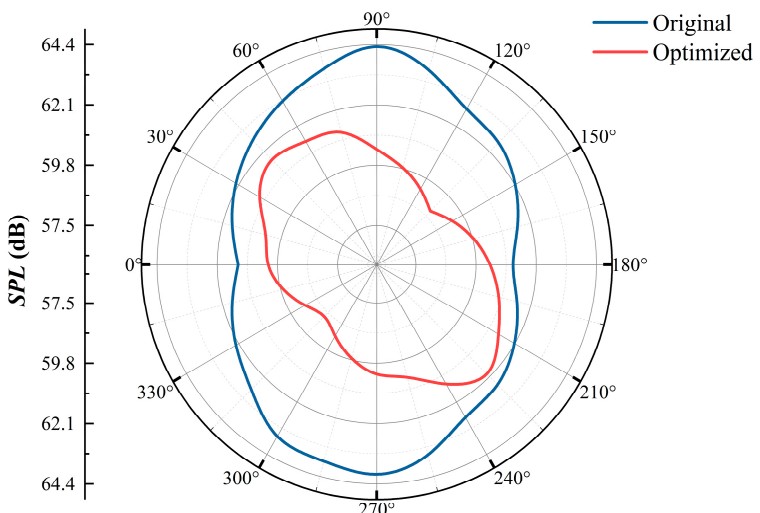

**Figure 25.** A circumferential noise pointing diagram.

## 6. Conclusions

This study is based on the multi-objective optimization of the rear guide vane of a diagonal flow fan under uncertain aerodynamic performance conditions. It verifies its feasibility and effectiveness through numerical simulations and experiments.

(1) The Pearson correlation analysis in the correlation analysis method is combined with the multi-objective optimization method to optimize the design of the rear guide vane of a small diagonal flow fan under uncertain aerodynamic performance conditions, reducing the influence of other parameters on the primary analysis parameters and showing the net correlation between each parameter. The total pressure is increased by 86 Pa, and the noise is reduced by 2.4 dB in the interval around the operating design point.

(2) The multi-objective robustness optimization design under aerodynamic performance uncertainty balances the traditional requirements of performance optimization and robustness design to ensure the robustness of performance while changing the mean value of aerodynamic performance to meet the requirements of fan performance under actual operating conditions. The optimization process is simple with few variables and has engineering application value, which can be extended to the optimal design of similar fluid machinery.

(3) The research in this paper provides insight into the characteristics of the flow field inside the rear guide vane of an inclined-flow fan under actual operating conditions. It optimises the blade shape of the rear guide vanes. The airflow velocity between the guide vanes is more uniform, thus significantly reducing flow losses in the fluid domain. The study provides a theoretical reference for improving the resistance to disturbances in the aerodynamic performance of the fan under uncertain conditions and ensuring the efficient, safe and stable operation of other impeller machinery such as the diagonal flow fan.

**Author Contributions:** Formal analysis, W.J.; Methodology, L.L.; Software, Z.M.; Writing—original draft, B.X.; Writing—review & editing, S.Z. All authors have read and agreed to the published version of the manuscript.

**Funding:** This research was funded by National Natural Science Foundation of China grant number 51706203 and Natural Science Foundation of Zhejiang Province grant number LY20E090004.

**Conflicts of Interest:** The authors declare no conflict of interest.

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
