# Peer review of "Multi-Objective Optimization of Rear Guide Vane of Diagonal Flow Fan Based on Robustness Design Theory"

_applsci, doi:10.3390/app12199858_

Round 1

Reviewer 1 Report

The authors claim to introduce "a variable-conditions, high-efficiency optimisation strategy" to find the best design parameters for the rear guide vane. The entire text needs to be rewritten and checked by a native-speaker as it is difficult to read or incomprehensible in places. CFD calculations were carried out using a commercial code. The entire paragraph 2.2 shows a lack of understanding of CFD and confusion of concepts, e.g. "The SIMPLE algorithm was chosen to ensure second-order accuracy" or "and the other terms in second-order windward format". The same is true for optimisation problems e.g. the sentence "The optimization mathematical model is optimised" suggests optimisation of the optimisation model (NSGA-II)! Because a multi-objective optimisation problem was considered, which means that a set of polypotimal solutions (Pareto front/set) is obtained, rather than the single solution in Figure 12. Where is the Pareto set and how was one solution selected from it?

In addition, the literature review and description of the current state of knowledge is insufficient and Figure 1 has no description in the text. In conclusion, I can say that the manuscript in its present form is not suitable for publication in any journal.

List of minor errors (just a few):
- page 1, 2: optimization vs optimise
- line 43: not "emory [4]" but "Emory [4]"
- line 44: not "using The uncertainty" but ". The uncertainty"
- line 55: what is "boundary speed"?
- lines 61, 62:  too many "Diagonal flow fan"
- line 82: not "r/min" but "rev/min" or "rpm"
- line 90: "Icem"? or "ICEM CFD"?
- line 100: "encrypted"?
- line 107: was it "TurboGrid" or "Icem"?
- line 114: "v is the kinematic viscosity", not "v" but Greek "nu"
- line 119: what is "constant numerical calculations"?
- line 122: "vortex viscosity" or "eddy viscosity"?
- line 140: and what is "uncertainty of the computational model" or what is "hybrid uncertainty, etc."?
- line 146: what is "cyclonic energy"?
- Formulas (2) and (3) are well known and there is no need to reproduce them here
- Where do the ranges of the variables in formula (4) come from?
- Figures 18, 19, 21 look like the results of a two-dimensional simulation

Author Response

Dear reviewers

Re: Manuscript ID: applsci-1924034 and Title: Multi-objective optimization of rear guide vane of diagonal flow fan based on robustness design theory

Thank you for your letter and the reviewers’ comments concerning our manuscript , those comments are valuable and very helpful. We have read through comments carefully and have made corrections. Based on the instructions provided in your letter, we uploaded the file of the revised manuscript. Revisions in the text are shown using red highlight for additions, and strikethrough font for deletions. The responses to the reviewer's comments are marked in red and presented following.

We would love to thank you for allowing us to resubmit a revised copy of the manuscript and we highly appreciate your time and consideration.

For a specific response, please see the attached document.

Best regards,

Biao Xu

Reviewer 2 Report

This paper presents a multi-objective optimization method based on robustness design theory. The followings should be carefully addressed in the revision to be published in applied sciences-Basel.

1.       For readers to quickly catch the paper’s contribution, it would be better to highlight major difficulties and authors’ original achievements to overcome them more straightforwardly in the abstract and introduction.

2.       Please carefully check the spelling and layout of the article (e.g., Line 112 ‘Y+’ and missing the third section).

3.       It is better to present the contours of the y+ distribution in Section 2.1.

4.       Please give details of the kriging model used in this paper.

5.       Qv=22m3/min,3 should be superscript. Z2 2 should be subscript. R2 value (0≤R2≤1)  2 should be superscript. Please check the gramme, format carefully.

6.       There is some Chinese in Fig. 10.

7.       How do the authors carry out the noise test? How to run the fan in fig. 16.

8.       Please give the Pareto front of the optimization. The authors just give the result in Table. 1

Author Response

(The authors gave the same response as above.)

Round 2

Reviewer 1 Report

1. I still thing that the literature review and description of the current state of knowledge is insufficient

2. line 84: "The network mainly consists of" or "The design mainly consists of"? Or perhaps "The arrangement mainly consists of", "The system mainly consists of".

3. line 104: The word "encryption" is not appropriate in this context and it is not clear what it means. I suggest removing the part "with encryption". The same applies to "and encrypted" in line 117.

4. line 107: "(a) Moving leaves" or "(a) Moving blades"?

5. line 115: "The dynamic vane area grid" or "The moving vane area grid"?

6. Line 118, 119: not "viscous bottom layer" but "viscous boundary layer"

7. line 125-126: not "used for the constant numerical calculations" but "used for the steady-state numerical calculations" I presume.

8. line 129-130: not "The eddy viscosity model uses the Realizable..." but "The turbulence modeling uses the Realizable..."

9. line 131-132: not "and the control equations were discretized" but "and the transport equations were discretized"

10. line 132-133: not "The second-order discrete format is chosen for the terms, and the second-order upwind design is selected for the other words." but rather something like this "The second-order accurate discretisation is chosen for individual terms and the second-order upwind is selected for the convection terms."

11. Line 152: "leaf shape"? do you mean "blade/vane shape"?

12. line 271, 289: "leaf height" or "blade/vane height"?

13. line 303: "leaf flow"?

14. line 311: "leaf channels"?

15. Line 343: "as shown in Fig." or "as shown in Fig. 23"?

16. line 368: "leaf shape" or "blade/vane shape"?

17. line 370: "leaf grille"?

Author Response

Dear reviewers

Re: Manuscript ID: applsci-1924034 and Title: Multi-objective optimization of rear guide vane of diagonal flow fan based on robustness design theory

Thank you for your letter and the reviewers' comments on our manuscript, which were very valuable and helpful. We have read these comments carefully and have made changes. Following the instructions in your letter, we have uploaded the revised manuscript file. Changes in the text are shown in blue to indicate additions. Responses to reviewers' comments are highlighted in red and are presented below.

We thank you very much for allowing us to resubmit a revised version of the manuscript, and we highly appreciate your time and consideration.

Best regards,

Biao Xu

Reviewer 2 Report

it can be accepted

Author Response

Dear reviewers

Re: Manuscript ID: applsci-1924034 and Title: Multi-objective optimization of rear guide vane of diagonal flow fan based on robustness design theory

Thank you for your letter and the reviewers' comments on our manuscript, which we have carefully read in full and revised. Following the instructions in your letter, we have uploaded a revised manuscript file. Modest English corrections have also been made, and changes in the text are indicated in blue as additions.

We thank you very much for allowing us to resubmit the revised manuscript, and we highly appreciate your time and consideration.

Best regards,

Biao Xu
